# Dive into the Scene: Autonomous Focus Plan Generation in Vision-Language Decision-Making

## Abstract

Vision-Language Models (VLMs) are promising in decision-making tasks, whereas the visual hallucination issue limits their performance in complex visual scenes. In such scenes, a number of visual objects exist, while the essential ones related to actions require focus in each step, avoiding the interference of unrelated objects. In this work, we propose SceneDiver, a coarse-to-fine, two-stage focus plan generation pipeline, to tackle the key technical challenge of identifying the essential objects from scenes with complicated visual and semantic structures. First, the VLM executes a virtual, coarse-grained plan over the scene graph. Then, it zooms into local neighborhoods around each graph node to perform fine-grained focusing. The resulting focus map controls the attentions of VLM during decision making, steering the model toward task-critical objects and alleviating the perceptual hallucination of VLMs. Experimental results under robotic manipulation and room navigation benchmarks demonstrate that our approach successfully overcomes the perceptual limitation of VLMs, meanwhile significantly enhances their decision-making performance and generalization ability. Our code will be open-released upon acceptance.

## 1 Introduction

Following the rapid advancement of vision-language reasoning ability for Vision-Language Models (VLMs), researchers have stepped forward to utilize VLMs in vision-language decision-making tasks (Zhai et al., 2024b). Similar to reasoning, decision-making also challenges the ability that VLMs correctly perceive and understand the visual scenes, while challenges even more on the ability of understanding how scenes change following the actions of the agent. Even though training specific vision-language-action models have become a frontier research topic (Ma et al., 2024), in this work, we focus on an alternative solution which can be meaningful in practice: Can we directly utilize off-the-shelf open/close-source VLMs without specific training on decision-making, even when their native multimodal perception and reasoning abilities are still limited?

Since the decision-making ability is built upon reasoning, it shares a similar unsolved issue for the reasoning ability of current VLMs, namely visual hallucination (Huang et al., 2024b), which means that VLMs may fall into the danger of inaccurately perceiving and understanding visual scenes, such as ignoring objects or identifying objects that do not actually exist. To tackle this fundamental challenge, existing work has proposed to break the visual perception process of VLMs into simpler steps to solve challenging visual reasoning tasks (Liu et al., 2024; OpenAI, 2025; Luo et al., 2025; Wu et al., 2025; Li et al., 2025a; Fu et al., 2025; Duan et al., 2025; Xu et al., 2025). The key idea lies in iteratively simplifying the visual scenes in each reasoning step via image modification operations, making the VLMs attend to key aspects related to the current reasoning step. Through this strategy, the visual hallucination issue is significantly alleviated without directly enhancing the native visual perception ability of VLMs. It is natural to think that the similar idea can also be utilized in decision-making, where the agent can utilize the simplified input visual scene to focus on essential aspects that are closely related to the choices of actions.

On the other hand, directly borrowing the existing techniques for multimodal reasoning is invalid in our scenario, since they either require VLM already has strong enough reasoning abilities to

autonomously conduct the essential scene modification operations, or require specific training under the task. To address this challenge, we propose SceneDiver, an effective method to boost the vision-language decision-making ability of VLMs. SceneDiver enables VLMs to autonomously generate the *focus plan*, which can help VLMs to focus only on tasks-relevant objects in each decision-making step. The central challenge lies in enabling VLMs to identify task-relevant objects from a cluttered scene containing numerous distractors. To address this, we introduce a coarse-to-fine, two-stage focus plan generation pipeline. At the first coarse-grained plan stage, we first convert raw image data into structured graph representations by constructing scene graphs. A virtual planning process is then performed over the graph, decomposing the complex global scene into a series of simpler local sub-scenes corresponding to individual nodes in the plan. At the second fine-grained plan stage, VLMs are requested to autonomously explore each local sub-scene using a set of naturally designed exploration strategies, ensuring that task-relevant objects are accurately discovered. Finally, the visual input is refined using the objects identified by the VLMs. Essential task-related information is preserved, while irrelevant content is suppressed. This refinement significantly improves the decision-making accuracy of VLMs.

We conduct experiments across diverse tasks of robotic manipulation (Feng et al., 2025) and room navigation (Yang et al., 2025) to verify the efficacy of our approach. The results show that Scene-Diver is able to achieve consistent improvements in the decision-making performance of VLMs. Furthermore, ablation studies indicate that iterative, step-by-step focusing outperforms one-shot focusing and further reduces noise introduced by scene graph. A sensitivity analysis of scene graph quality shows that our approach is robust: even with untrained scene graph modules, focus planning still yields measurable accuracy gains. Overall, structured and incremental attention control offers a practical route to alleviating multi-object hallucination and improving the dependability of VLM-based decision making.

In summary, our contributions are listed as follows:

- To tackle the challenge of utilizing the limited native perception and reasoning abilities of VLMs in decision-making without specific training, we propose an autonomous paradigm of focus plan generation, enabling reducing the hardness of decision-making in each step.
- We propose SceneDiver, a novel method to enable VLMs to autonomously generate the focus plan under the coarse-to-fine paradigm. Aided by both semantic and spatial information of the visual scene, SceneDiver enables VLMs to autonomously construct a focus plan over essential objects, which then modifies the input image to suppress distracting content and guides the decision-making process.
- We conduct systematic experimental analysis to justify the efficacy of our approach. The results soundly verify the rationality of the proposed focus plan generation paradigm and the strong performance of SceneDiver in improving the vision-language decision-making abilities of various base VLMs.

## 2 RELATED WORK

### 2.1 VISION-LANGUAGE DECISION-MAKING

Vision-language decision-making has become a promising research area (Ma et al., 2024). Benefiting from the perceptual capabilities of MLLMs and their comprehension of natural language instructions, Traditional control policies have become more diversified and are now capable of interacting with users in a more intuitive manner. For instance, (Brohan et al., 2023) employs MLLMs as high-level task planners, enabling the decomposition of user-provided instructions into a sequence of plausible low-level skills. Similarly, (Huang et al., 2022) adopts a two-stage framework to translate high-level instructions into executable actions. (Jang et al., 2022) achieves zero-shot task generalization to unseen tasks by aligning language instructions or human demonstration videos with visual observations of the environment. The performance of decision-making in such systems is closely tied to the effectiveness of the vision encoder. Accordingly, numerous approaches have been proposed to enhance the visual processing components of MLLMs. (Shang et al., 2024) integrates knowledge distilled from diverse vision foundation models into a unified architecture to maximize visual representation capacity. (Assran et al., 2023) constructs an implicit world model by comparing embeddings of image patches to capture structural regularities. (Karamcheti et al., 2023) further

improves vision-language alignment by incorporating language conditioning and generation into the masked image autoencoder (MAE) training objective.

## 2.2 OBJECT HALLUCINATION

Despite VLM's promising performance on benchmarks (Danish et al., 2025; Li et al., 2025b; Zong et al., 2024), these models (Liu et al., 2023; Chen et al., 2024a) frequently generate objects that do not exist in the provided images, a problem known as object hallucination (Rohrbach et al., 2019; Dai et al., 2023). Prior work has explored several avenues to mitigate this issue, including integrating an external object detector (Zhai et al., 2024a), applying visually grounded visual instruction tuning (You et al., 2023; Zhang et al., 2024) or using reinforcement learning (Sun et al., 2023; Gunjal et al., 2024).

While prior work focuses on general object hallucination, the specific challenge of multi-object hallucination—where models invent multiple entities with incorrect attributes and relationships—remains less explored. Chen et al. (2025) finds that high cognitive loads from multi-object queries lead models to use heuristic shortcuts, bypassing rigorous visual analysis for each object. To mitigate this, researchers explore several avenues. Zhu et al. (2025) identifies that non-uniform spatial attention causes identical objects to be processed differently based on their position, and proposes a training-free attention rectification method to address this. Targeting a phenomenon termed "local over-trust", Huang et al. (2024a) develops a method that monitors attention weights during decoding, which penalizes generation paths where a single token excessively influences subsequent content.

## 3 METHOD

### 3.1 OVERVIEW

We view vision-language decision-making as a mapping conditioned on the visual observation and the language-based task instruction. Let $o$ denote the encoded visual observation and $q$ the instruction; the VLM outputs an action $a = \pi(o, q)$. To characterize observation quality, we decompose $o$ into task-relevant visual evidence $o_{\text{rel}}$, task-irrelevant/distracting information $o_{\text{irr}}$, and model-induced perceptual hallucinations $o_{\text{hall}}$. Under this formulation, $o_{\text{rel}}$ improves decision accuracy, whereas $o_{\text{irr}}$ and $o_{\text{hall}}$ degrade it. In particular, hallucination of VLMs could introduce multiple spurious objects or attributes, causing systematic mis-grounding and attention shifts that ultimately reduce decision accuracy.

VLMs frequently hallucinate multiple objects in cluttered scenes due to object-level interference (Chen et al., 2024b), which in turn misguides downstream decisions. To mitigate this, we boost decision-making by letting the VLM autonomously generate a focus plan and by editing the image to suppress spurious content. Specifically, we adopt a coarse-to-fine two-stage focus: In the first stage, a virtual plan is executed on the scene graph, serving only to reveal critical objects that may influence decision-making. In the second stage, we treat each node of the virtual plan as an anchor point and apply more fine-grained focusing to objects within its local neighborhood. Together, the two-stage focusing yields a pixel-level focus map, which we use to modify the input image and steer the model's attention toward essential targets.

### 3.2 STAGE ONE: COARSE-GRAINED PLAN OVER THE SCENE GRAPH

In this stage, we aggregate scene information at a global level to obtain a coarse estimate of the targets to be focused. Concretely, we construct a scene graph that decomposes the environment into object nodes and relational edges, and instruct the VLM to execute a lightweight virtual plan over this structure, yielding an interpretable trajectory and a set of anchor nodes. These anchors serve two purposes: (i) they act as prioritized candidates for subsequent fine-grained focusing, substantially shrinking the search space; and (ii) they enable structural validation of candidates via relational consistency and graph connectivity, thereby mitigating multi-object hallucination and duplicate detections.

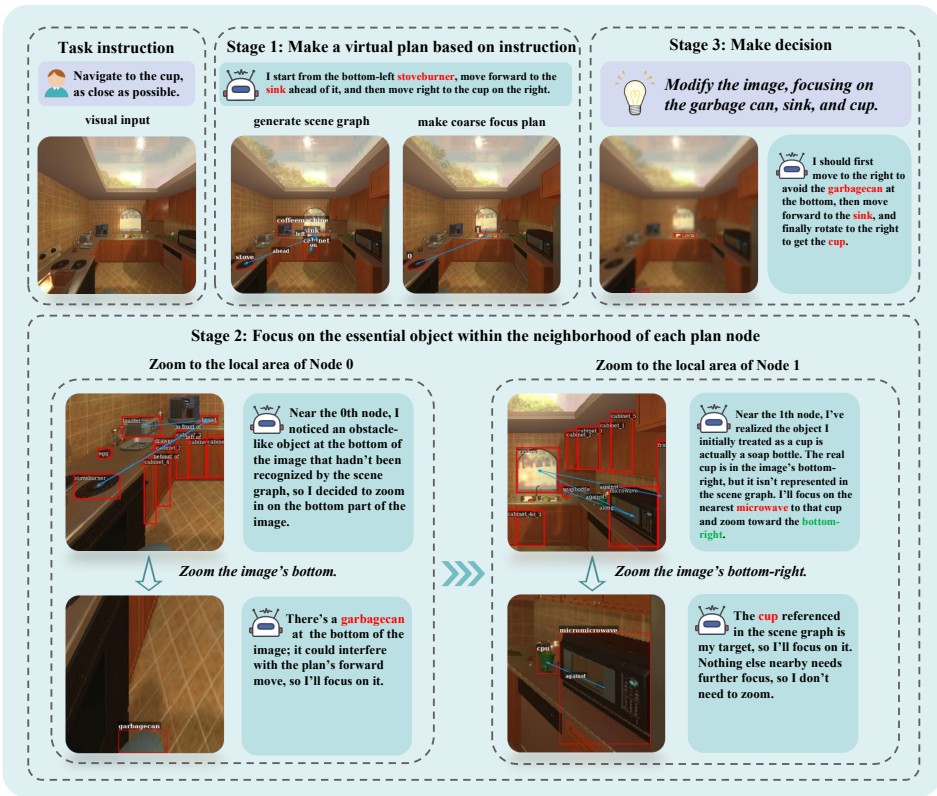

Figure 1: Overview of SceneDiver: From the input image we build a scene graph, generate a virtual plan (Stage 1), perform local focusing around each plan node to identify essential objects (Stage 2), and use the resulting focus to modify the image and make decision.

We utilize OvSGTR Chen et al. (2024d) to extract information from images and construct scene graphs. Detected objects are annotated with <ref>, and their spatial relationships are specified by <pred>. The location of each object is encoded using <box>. Confidence scores, denoted as <conf>, are assigned to both objects and relationships to quantify reliability.

To generate the virtual plan, the VLM reasons over the scene graph—using its object nodes and relational edges—to infer the current and goal states, and selects the necessary intermediate nodes across objects, yielding a lightweight planning trajectory. To ensure object consistency and verifiability, the model must explicitly reference each target with the <ref> (object identifier) and <box> (spatial coordinates) tags when describing its semantics and location. These structured cues are cross-checked against the scene graph to verify actual existence and promptly correct hallucinations. By enforcing <ref>/<box> alignment between detection outputs and textual references, the approach reduces ambiguity in multi-instance scenes and improves the consistency of downstream reasoning.

### 3.3 STAGE TWO: FINE-GRAINED FOCUSING IN LOCAL NEIGHBORHOODS

The coarse virtual plan provides a high-level route that is subsequently refined through a localized, iterative reasoning process. In the fine-grained stage, the VLM sequentially focuses on each node of the coarse plan, utilizing a limited perception field. This methodology is a training-free mechanism that reduces distractions from irrelevant visual information. This focus has a dual benefit: it lessens the likelihood of model hallucination and simultaneously directs the model's attention to perceive fine-grained details.

This refinement phase intentionally restores the model's autonomy, which was constrained during the initial high-level planning. By granting this freedom only when the model is reasoning over a

limited perceptual field, its decisions become more grounded and reliable. This approach aids the VLM in identifying and correcting potential inaccuracies made during the coarse planning stage. To this end, the VLM exercises its autonomy at each node by reflecting on the plan and choosing from three straightforward strategies: 1) **follow the plan** along the suggested path, 2) **look closer** to gather more detailed visual information about the immediate surroundings, or 3) **look outwards** to broaden its view and re-contextualize its current position.

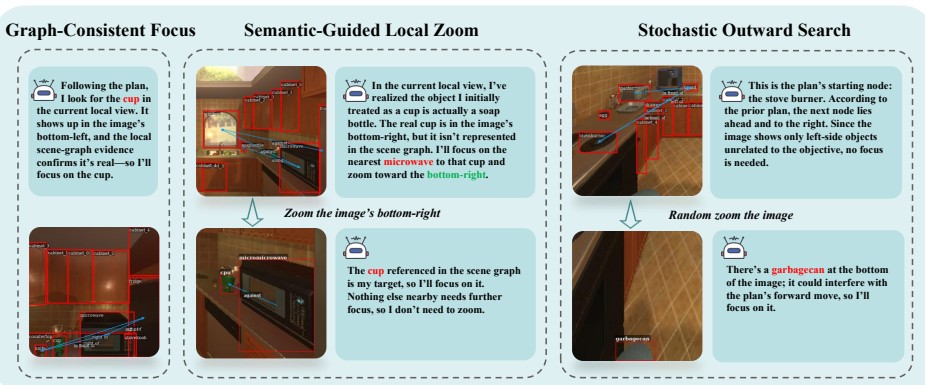

Figure 2: Three straightforward strategy examples.

Specifically, during the fine-grained focusing stage, we maintain a candidate set $\mathcal{C}$ of essential objects to be attended. The process begins at the first node $n_0$ on the plan path $\pi$. At step $t$, attention is restricted to a local window $W_t$ centered at the current node $n_t$. Depending on observations, one of three strategies is invoked:

1. **Graph-Consistent Focus.** If the object currently attended by the planner is *consistent with the plan path* (i.e., it is detected and matches an indexed node or its vicinity), it is directly added to $\mathcal{C}$. The system then performs a zoom operation oriented toward the next node $n_{t+1}$ on the guidance path $\pi$.

2. **Semantic-Guided Local Zoom.** If the attended object cannot be reliably matched to the scene graph due to planner hallucination or insufficient local resolution, a *local, more fine-grained zoom* is executed based on the planner's semantic or position description. This reduces surrounding clutter while enlarging the candidate target. Once a valid candidate is confirmed or a predefined depth limit is reached, the procedure returns to the anchor point and a zoom is directed toward the next node on $\pi$.

3. **Stochastic Outward Search.** If the planner cannot localize a plausible target, a *stochastic outward search* is performed in multiple directions around $n_t$. The search terminates when the maximum exploration budget is exhausted, or as soon as either of the above conditions is satisfied.

### 3.4 GUIDING REAL PLAN THROUGH FOCUS SCORE MAP

In this section, we introduce an explicit representation built from essential-object candidate set. By solely modifying the input image, this method directs the planner's attention to essential objects without any internal model changes.

#### 3.4.1 FOCUS SCORE MAP CONSTRUCTION

We construct a two–dimensional attention map $A \in \mathbb{R}^{H \times W}$ with the same spatial resolution as the input image, where each entry encodes a per–pixel attention score. Let $\mathcal{C}$ denote the set of candidate objects. For each candidate object $i \in \mathcal{C}$, we assign its region a focus-map score based solely on its distance to the image center. Let $p_i = (c_x^{(i)}, c_y^{(i)})$ be the region midpoint and $(x_c, y_c)$ the visual center; let $d_i$ denote the Euclidean distance from $p_i$ to $(x_c, y_c)$, and $d_{\max}$ the distance from $(x_c, y_c)$

Table 1: Robot manipulation accuracy. Variances over 5 seeds are reported in the parentheses.

| Model | Base | Focus |
|---|---|---|
| Qwen2.5-VL-7B-AWQ (Bai et al., 2025) | 0.1467(0.0016) | **0.2867(0.0012)** |
| Qwen2.5-VL-32B-AWQ (Bai et al., 2025) | 0.2133(0.0007) | **0.3133(0.0003)** |
| gpt-4o-mini (OpenAI, 2024) | 0.2867(0.0020) | **0.3400(0.0024)** |
| gemini-2.5-flash-nothinking (Comanici & et al., 2025) | 0.3867(0.0025) | **0.4667(0.0018)** |

to the farthest image corner. The coefficient is

$$c(d) = \exp\left(-\tfrac{1}{2}\left(\frac{d}{\sigma \, d_{\max}}\right)^2\right).$$ (1)

which mirrors human visual attention—items closer to the center naturally attract stronger focus, and the coefficient increases as distance decreases.

### 3.4.2 RELEVANCE-GUIDED IMAGE EDITING VIA FOCUS SCORE MAP

Let $I \in [0,1]^{H \times W \times C}$ be the input image and $s \in [0,1]^{H \times W}$ a normalized per-pixel score map. We reweight the image pixelwise to emphasize high-score regions while keeping low-score areas slightly dimmed instead of hard-masked, thereby preserving contextual information that benefits planning tasks. To avoid blacking out low-score regions, we use a floor parameter $\beta$:

$$I' \,=\, I \odot \big(\beta + (1-\beta)\,s\big),$$ (2)

where $\odot$ denotes element-wise multiplication.

Further de-emphasis of low-score areas can be obtained without introducing hard boundaries by deriving a spatially varying blur strength from $(1-s)$. Let

$$a \,=\, G_\sigma * (1-s),$$ (3)

where $G_\sigma$ is a Gaussian smoothing operator applied to the scalar field. With a Gaussian image blur $\mathcal{B}_{\sigma_b}$, we synthesize a soft-focus composite:

$$I'' \,=\, (1-a) \odot I' \,+\, a \odot \mathcal{B}_{\sigma_b}(I),$$ (4)

which keeps salient content sharp and bright while gently defocusing background.

## 4 EXPERIMENT

We conduct experiments on two distinct benchmarks. The first benchmark quantifies both end-task performance and the accuracy of our focus procedure via oracle annotations, allowing us to isolate the contribution of correct focus from downstream planning. The second benchmark evaluates robustness and generality across a range of open- and closed-source models; we also perform ablations on each component to substantiate our architectural decisions. Additionally, we study the sensitivity of our method to the underlying scene-graph pipeline, characterizing how scene-graph quality propagates to the focus mechanism and ultimately impacts decision-making performance.

### 4.1 ROBOT MANIPULATION

In the first experiment, we constructed a robotic-arm task in MuJoCo (Todorov et al., 2012) following the setup of Feng et al. (2025). The scene consists of a base plate with small pieces randomly scattered across a tabletop, augmented by a decoy board and several decoy pieces that serve as distractors. The objective is to assemble the pieces onto the base plate, one by one. The VLM agent controls the robotic arm to manipulate the bricks using four actions: *pick up*, *insert*, *reorient*, and *put down*.

Figure 3: Decision-making trajectory for robot manipulation.

To evaluate our method, we randomly sample 30 distinct scenes to evaluate VLM performance. In each scene, the VLM receives an image of the current state and an image of the target state, and must complete the assembly within thirty steps. We report the success rate across the 30 scenes and repeat the experiment five times; Table 1 summarizes the mean and variance over the five runs. The results demonstrate that our method performs well across multiple models and effectively improves the decision accuracy of VLMs in complex environments.

We illustrate a concrete decision trajectory in Figure 4.1. The red boxes are for visualization only and are not visible to the VLM. We present the processing at step 4, which shows that the VLM recognizes that the current goal is to assemble the green block by constructs a virtual plan over the scene graph. However the initial plan ignore the block's orientation, after applying a fine-grained local neighbor focus, the VLM correctly attends to the orientation issue and to the cradle that enables reorienting the block. Aggregating essential objects across multiple focus steps yields the final focused plan. We then modify the image accordingly to condition the next decision, which ultimately leads to the successful assembly of the green block.

## 4.2 ROOM NAVIGATION

Building upon the room-navigation benchmark by Yang et al. (2025), we propose a more demanding variant designed to push the limits of decision making. We retain the original protocol where the agent navigates using only visual observations and environmental feedback, but introduce two key challenges. First, we select scenes with substantially higher environmental complexity. Second, we make target objects harder to identify by making them smaller, more frequently occluded, and easily confusable with visually similar distractors, further increasing perceptual ambiguity.

We evaluate our method on four sub-tasks. The **base** sub-task uses the shortest, most direct instructions; the **common-sense** sub-task presents instructions in everyday conversational form; the **complex-instruction** sub-task requires the VLM to identify the target within a lengthy, intricate prompt; and the **visual-appearance** sub-task specifies the target indirectly via its shape and color rather than by name. We assess both open-source and closed-source models on each sub-task, running five independent trials per task. Table 2 reports the mean and variance of decision accuracy across the five runs.

As shown in Table 2, compared with directly letting the VLM make decisions (the base model), our method effectively improves decision accuracy. To assess the influence of each component and substantiate our design choices, we conduct three ablation experiments.

**Graph.** In our approach, the scene graph serves as structured scene knowledge. To assess whether scene graphs alone can improve the decision-making of VLMs, we feed the textualized scene graph to the models as auxiliary input. The results show that scene-graph information can help VLMs to some extent, but complex scenes make the graphs highly verbose. When the scene-graph text becomes lengthy, it draws attention away from the image and exacerbates hallucinations. Moreover, noise introduced during scene-graph construction can easily mislead the models to focus on spu-

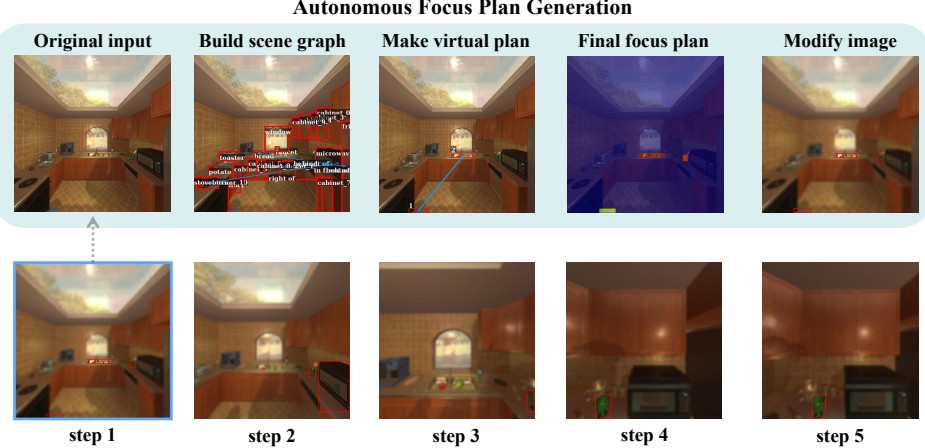

Figure 4: Decision-making trajectory for room navigation.

Table 2: Quantitative comparisons under the room navigation benchmark. Variances over 5 seeds are reported in the parentheses.

| Subtasks | Base Model | Graph | Direct Focus | Only Coarse Focus | Ours |
|---|---|---|---|---|---|
| **Qwen2.5-VL-7B-Instruct-AWQ** (Bai et al., 2025) | | | | | |
| Base | 0.3267(0.0011) | 0.3067(0.0006) | 0.3400(0.0024) | 0.3733(0.0002) | **0.4400(0.0006)** |
| Common Sense | 0.3067(0.0002) | 0.2800(0.0003) | 0.3133(0.0003) | 0.3133(0.0007) | **0.3600(0.0002)** |
| Complex Instruction | 0.3200(0.0007) | 0.3467(0.0016) | 0.3267(0.0002) | 0.3467(0.0003) | **0.3733(0.0002)** |
| Visual Appearance | 0.2733(0.0028) | 0.3133(0.0012) | 0.3000(0.0000) | 0.3200(0.0012) | **0.3533(0.0003)** |
| **Qwen2.5-VL-32B-Instruct-AWQ** (Bai et al., 2025) | | | | | |
| Base | 0.4933(0.0006) | 0.5067(0.0002) | 0.5133(0.0007) | 0.5200(0.0003) | **0.5667(0.0009)** |
| Common Sense | 0.4333(0.0018) | 0.4533(0.0012) | 0.4667(0.0004) | 0.4667(0.0013) | **0.5267(0.0006)** |
| Complex Instruction | 0.4667(0.0009) | 0.4933(0.0015) | 0.4800(0.0003) | 0.4867(0.0016) | **0.5333(0.0004)** |
| Visual Appearance | 0.4333(0.0004) | 0.4533(0.0003) | 0.4533(0.0003) | 0.4600(0.0002) | **0.4933(0.0011)** |
| **InternVL2.5-8B** (Chen et al., 2024c) | | | | | |
| Base | 0.2933(0.0006) | 0.3133(0.0003) | 0.3267(0.0002) | 0.3733(0.0002) | **0.3933(0.0011)** |
| Common Sense | 0.2267(0.0006) | 0.2533(0.0012) | 0.2733(0.0006) | 0.2867(0.0016) | **0.3267(0.0002)** |
| Complex Instruction | 0.2400(0.0006) | 0.2600(0.0011) | 0.2867(0.0003) | 0.2733(0.0011) | **0.3333(0.0009)** |
| Visual Appearance | 0.2133(0.0012) | 0.2333(0.0004) | 0.2267(0.0006) | 0.2333(0.0009) | **0.2533(0.0003)** |
| **gpt-4o-mini** (OpenAI, 2024) | | | | | |
| Base | 0.4333(0.0013) | 0.4400(0.0011) | 0.4400(0.0011) | 0.4600(0.0006) | **0.5000(0.0004)** |
| Common Sense | 0.3733(0.0028) | 0.3867(0.0020) | 0.4400(0.0006) | 0.4733(0.0002) | **0.5333(0.0018)** |
| Complex Instruction | 0.3867(0.0012) | 0.4067(0.0006) | 0.4467(0.0003) | 0.4267(0.0002) | **0.4933(0.0006)** |
| Visual Appearance | 0.3733(0.0015) | 0.3933(0.0006) | 0.4267(0.0002) | 0.4400(0.0006) | **0.4933(0.0015)** |
| **gemini-2.5-flash-nothinking** (Comanici & et al., 2025) | | | | | |
| Base | 0.6800(0.0007) | 0.6867(0.0020) | 0.7067(0.0002) | 0.7267(0.0002) | **0.7467(0.0007)** |
| Common Sense | 0.6200(0.0003) | 0.6267(0.0002) | 0.6400(0.0002) | 0.6333(0.0004) | **0.6533(0.0003)** |
| Complex Instruction | 0.6000(0.0004) | 0.6133(0.0003) | 0.6333(0.0018) | 0.6333(0.0022) | **0.6600(0.0011)** |
| Visual Appearance | 0.5533(0.0007) | 0.5600(0.0006) | 0.5867(0.0003) | 0.5733(0.0015) | **0.6200(0.0016)** |
| **doubao-seed-1.6-flash-nothinking** (Guo & et al., 2025) | | | | | |
| Base | 0.5933(0.0006) | 0.6067(0.0002) | 0.6267(0.0006) | 0.6333(0.0009) | **0.6600(0.0006)** |
| Common Sense | 0.4000(0.0009) | 0.4133(0.0003) | 0.4400(0.0006) | 0.4600(0.0006) | **0.4800(0.0003)** |
| Complex Instruction | 0.3800(0.0025) | 0.3933(0.0011) | 0.4133(0.0016) | 0.4067(0.0015) | **0.4867(0.0007)** |
| Visual Appearance | 0.3267(0.0011) | 0.3600(0.0011) | 0.4000(0.0013) | 0.4067(0.0011) | **0.4467(0.0003)** |

rious objects during decision making. Consequently, while using scene graphs alone can improve decision-making performance, this approach suffers from limited robustness.

**Direct Focus.** To test whether a step-by-step focusing strategy outperforms a direct-focus approach, we supply the VLMs with the same scene-graph data as in our method. We then let the models directly output the essential object to focus on, and process the image accordingly using the same

downstream pipeline. The results show that this direct-focus variant can also improves decision accuracy and is more stable than feeding the scene graph as raw text. However, it lacks control over global scene context: the VLM tends to fixate on the final target while ignoring the current state and intermediate constraints. In tasks like visual appearance, the model is easily distracted by look-alike objects and fails to reach a globally optimal decision. Instead, our method performs step-by-step focusing via a virtual plan, which reduces the occurrence of these failures.

**Only Coarse Focus.** Beyond global control, our method applies fine-grained focus at each node of the plan. To assess its necessity, we evaluate decision accuracy with coarse-grained focus only. The results show that, given the constraints of the current state, focusing only on the coarse plan leads to erroneous predictions of future states, which in turn introduces hallucinated objects into the plan. Meanwhile, enforcing fine-grained focus at every node effectively mitigates these hallucinations and improves the robustness of the method.

In our method, the scene graph plays a central role. We seek to achieve the best performance under a minimal training budget for the scene-graph generator. Accordingly, we ablate which components of the generator to train with limited supervision. The configurations are: (1) No training (zero-shot generalization); (2) Detector-only training; and (3) Detector and relation predictor training. This design reveals how much each component contributes when training is scarce and identifies the most cost-effective strategy.

As shown in Fig. 5, even without training the scene-graph generator, our method consistently improves the VLMs' decision-making performance, owing to the strong out-of-the-box generalization of OvS-GTR. When we fine-tune the detection head, the scene graph recovers more small or occluded objects; this not only boosts accuracy but also reduces the number of zoom steps by

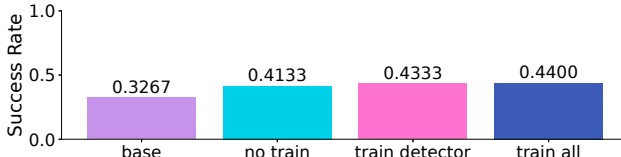

Figure 5: Qualitative comparison of SGG training.

identifying targets earlier. On the other hand, training the relation predictor primarily strengthens VLM spatial reasoning, providing greater gains on tasks dependent on inter-object relations.

## 5 CONCLUSION

We propose an autonomous focusing paradigm to mitigate multi-object hallucination in vision–language models (VLMs) during decision-making. Our method enables VLMs to autonomously construct a focus plan over essential objects, while modifying the input image to suppress distracting content and guide the model's decision making. Experimental results on multiple tasks show consistent improvements in VLM decision accuracy. Ablation studies reveal that iterative, step-by-step focusing consistently outperforms one-shot focusing, further mitigating noise introduced by the scene graph. Sensitivity analysis of scene graph quality demonstrates the robustness of our approach: even with untrained modules, focus planning delivers measurable accuracy gains. Overall, structured and incremental attention control provides a practical means to reduce multi-object hallucination and enhance the reliability of VLM-based decision making.

**Limitations and future work**. We discuss three limitations for our current approach. First, the current approach focuses on the visual modality, targeting at addressing visual hallucinations only. Since tackling the hallucination issue is also an unaddressed challenge for textual reasoning, future work can further utilize the learning to focus paradigm under multi-modalities. Second, our current approach utilizes relatively simple inpainting strategy to remove the less important objects in the scenes since we explore more on the challenge of deciding *which and where to focus* than *how to focus* in this work. While future research can indeed utilize our approach with stronger inpainting strategy to address more complicated real-world applications. We also believe that extending our task to 3D scenes can also be an interesting problem, where 3D modeling techniques allows the focus operation to be easier to realize, as done in Liu et al. (2024). In the last, our current approach assumes to utilize pre-trained vision-language decision-making agents. It is also interesting to study whether the ability of focusing can be obtained jointly with decision-making by learning from scratch.

REPRODUCIBILITY STATEMENT

We will open-release the code upon acceptance, which will serve as the reliable resource to fully reproduce our method and results.

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

## A ADDITIONAL EXPERIMENTAL RESULTS

### A.1 ROOM NAVIGATION

In this section, we present detailed results for the navigation task across its subtasks. Each subtask uses a distinct instruction to probe different facets of the model's decision-making. For every subtask, we provide two decision trajectories along with the corresponding instruction. For readability, we highlight the focused targets with red bounding boxes. Note that all annotations are invisible to the VLMs.

**Base.** In this subtask, the instruction is very explicit, for example: "navigate to the pot in the room and get as close as possible to it." The VLMs can directly observe the target object in the scene.

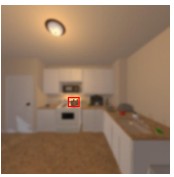 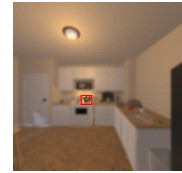 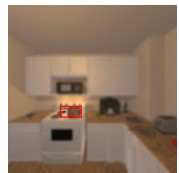 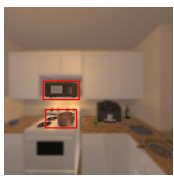 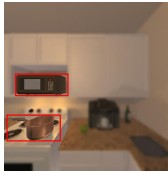

Figure 6: Instruction: navigate to the Pot in the room and be as close as possible to it.

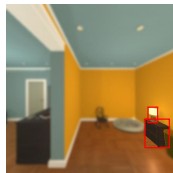 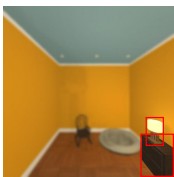 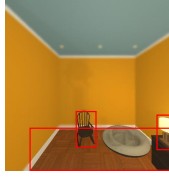 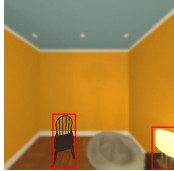 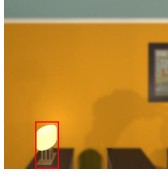

Figure 7: Instruction: navigate to the DeskLamp in the room and be as close as possible to it.

**Common Sense.** Compared to the base task, this task involves a more complex instruction, requiring stronger reasoning capabilities from the MLLMs.

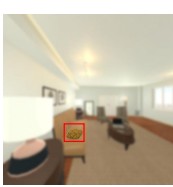 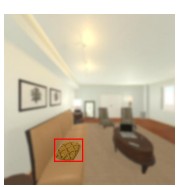 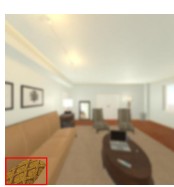 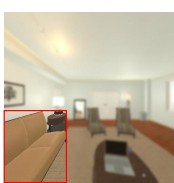 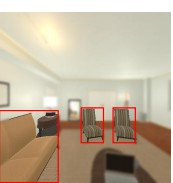

Figure 8: Instruction: I need a soft cushion to support my head while sleeping. Can you navigate to that object and stay close?

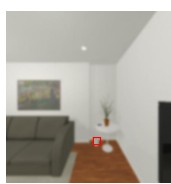 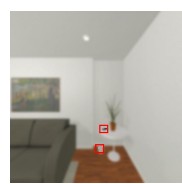 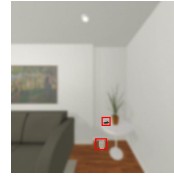 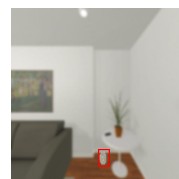 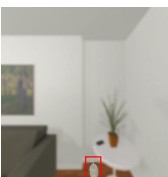

Figure 9: Instruction: I'd like to view a decorative sculpture representing a figure or person. Can you navigate to that object and stay close?

**Complex Instruction.** In this task, the instructions often contain substantial content irrelevant to the target, requiring VLMs to identify the task objective within complex prompts and align it with the objects in the image.

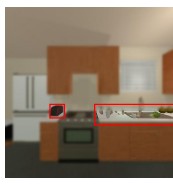 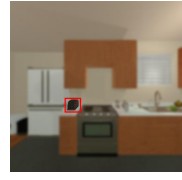 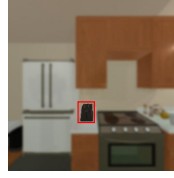 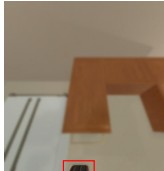 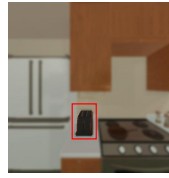

Figure 10: Instruction: The sound of someone walking upstairs adds a subtle rhythm to the quiet morning. There's a folded towel on the counter, and the air smells faintly of butter. Could you navigate to the toaster for me? It's a peaceful start to the day.

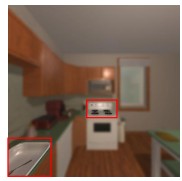 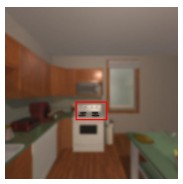 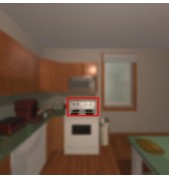 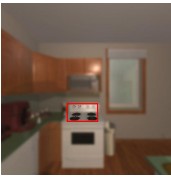 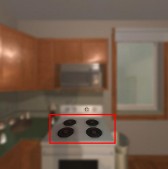

Figure 11: Instruction: The rhythmic ticking of the kitchen clock blends with the occasional drip from the faucet. There's a small pile of onions on the table, freshly chopped. Please move towards the stove burner for me. The kitchen has a comforting hum to it.

**Visual Appearance Task.** The instruction in this task describes the visual appearance of the target object, requiring the MLLMs to possess strong visual comprehension capabilities.

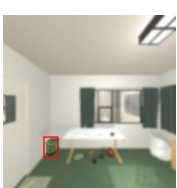 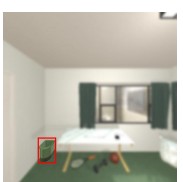 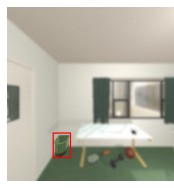 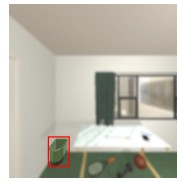 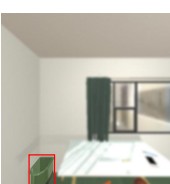

Figure 12: Instruction: Approach the tall green container with a smooth texture.

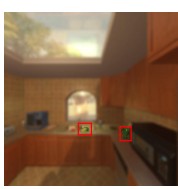 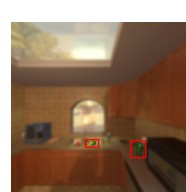 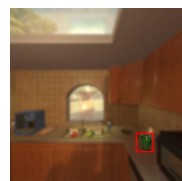 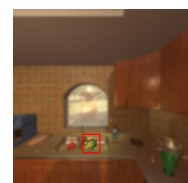 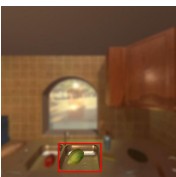

Figure 13: Instruction: Move closer to the small round object with a green surface and a cylindrical shape.

## B  IMPLEMENTATION DETAILS

In this section, we provide a detailed description of the implementation of our proposed method.

### B.1  EXPERIMENT SETTING

Our navigation is based on Yang et al. (2025) room navigation tasks. On this basis, we modify the target objects to be more challenging and more likely to distract the decision-making of VLMs. We

instantiate an AI2-THOR controller with cloud rendering and standardized perceptual parameters: grid size $g = 0.1$ m, visibility distance $d_{\mathrm{vis}} = 10$ m, field of view FoV $= 100°$, and image resolution $500 \times 500$ px, with depth and instance segmentation disabled. Let $\mathcal{S}$ denote the ordered set of scenes obtained from the input mapping; for each $s \in \mathcal{S}$, the environment is reset and the agent's initial ground-plane position $(a_x, a_z)$ is read from metadata.

Each object $o$ exposes an axis-aligned bounding box (AABB) with side lengths $(s_x, s_y, s_z)$ and a ground-plane center projection $(c_x, c_z)$. We define the proxy volume

$$V(o) \;=\; \max(s_x, 0)\, \max(s_y, 0)\, \max(s_z, 0).$$

An object is considered *small* if and only if it is visible, it is not one of the excluded planar structures, and its volume is strictly below a tunable threshold:

$$\mathrm{visible}(o) = \mathrm{True}, \quad \mathrm{objectType}(o) \notin \{\mathrm{Floor, Wall, Ceiling, DoorFrame, Window}\}, \quad 0 < V(o) < \tau,$$

with default $\tau = 0.05$ m$^3$. This filter biases the target set toward visually subtle, manipulation-scale items and excludes trivial background geometry.

To raise navigational difficulty, we impose a minimum agent–target separation in the $x - z$ plane. Let
$$D((a_x, a_z), (c_x, c_z)) = \sqrt{(a_x - c_x)^2 + (a_z - c_z)^2}$$
be the Euclidean distance; admissible targets must satisfy

$$D((a_x, a_z), (c_x, c_z)) \geq \delta,$$

where $\delta \geq 0$ is a hyperparameter representing the minimum distance. Larger $\delta$ systematically enforces longer trajectories and harder exploration before interaction.

Selection becomes adversarial with respect to distance: among candidates that pass the volume and distance filters, we choose

$$o^\star \;=\; \arg\max_{o \in \mathcal{C}} D((a_x, a_z), (c_x(o), c_z(o))),$$

where $\mathcal{C}$ is the admissible set. Otherwise, we sample uniformly at random from $\mathcal{C}$, preserving diversity while retaining the same geometric constraints.

If no candidate satisfies the smallness and distance criteria ($\mathcal{C} = \varnothing$), we fall back to a visible, pickupable set to preserve scene coverage, sampling uniformly:

$$o^\dagger \sim \mathrm{Unif}\big(\{o: \ \mathrm{visible}(o) \wedge \mathrm{pickupable}(o)\}\big).$$

This fallback weakens difficulty but guarantees that each scene yields a target type.

For each scene $s$, we record only the object type of the selected target, producing a mapping $M: s \mapsto \mathrm{objectType}(o)$. Stochasticity is controlled by a global PRNG seed $r$, enabling exact replication of scene-level assignments under fixed $\tau, \delta$.

The setting exposes clear difficulty knobs and their effects: decreasing the volume threshold $\tau$ contracts candidates to smaller items, increasing perceptual difficulty (reduced pixel footprint and increased clutter sensitivity); increasing the minimum distance $\delta$ raises exploration and path-planning load; enabling the farthest-target policy maximizes path length within scene constraints; varying the seed $r$ yields alternative hard target placements for robustness analysis while keeping comparability under a fixed seed.

In summary, by jointly constraining geometric scale ($V < \tau$), enforcing spatial separation ($D \geq \delta$), and optionally maximizing distance during target selection, the protocol increases perceptual, navigational, and search difficulty while remaining reproducible and coverage-preserving. This yields a standardized, tunable hard regime for embodied small-object seeking in AI2-THOR.

### B.2 SceneGraphGeneration

We adopt the OvSGTR algorithm proposed by Chen et al. (2024d) as our scene-graph generator. Thanks to its strong inherent generalization, effective training requires only 200 samples. Our training set is synthesized via randomly generated scenes: for each scene, we read the 3D coordinates

and semantic labels, then project the 3D points to 2D using the camera pose. Finally, we train the model with the official OvSGTR training scripts.

To train OvSGTR, We develop a synthetic data generator in AI2-THOR to train both the detector and the predicate predictor of scene graph generation models. The generator operates across multiple indoor scenes and, for each scene, performs short stochastic explorations to acquire diverse viewpoints. Rendering uses a cloud pipeline with depth and instance segmentation enabled, a square viewport of 500×500 pixels, a 100° field of view, a 0.1 m grid step, and a 10 m visibility distance. All randomness is seeded to ensure reproducibility.

Within each scene, the agent follows a randomized walk that samples exactly 30 frames. Each iteration first records the current viewpoint and then executes a single action uniformly drawn from a compact action set: translations of 0.25 m (forward, back, right, left), yaw rotations of 90° (left or right), and pitch adjustments of 30° (up or down). If a translation fails due to collisions or constraints, the policy triggers a fallback yaw of 45°, 60°, or 90°, maintaining smooth exploration without relying on precomputed reachable points. This lightweight control strategy yields broad coverage of object configurations while remaining computationally frugal.

Each frame provides three synchronized products suitable for scene graph supervision. First, an RGB image at 500×500 resolution. Second, an instance index map (with per-instance boolean masks if available, or masks reconstructed from the index map otherwise). Third, OvSGTR/VG-style annotations capturing object nodes and pairwise relations. Visible objects are aligned to engine metadata via unique identifiers, their category names are normalized (underscores and hyphens mapped to spaces, lowercased), and compact 2D bounding boxes are computed from the corresponding instance masks, discarding degenerate cases.

To support geometry-aware relations, the generator retains, for each object, axis-aligned 3D descriptors (center, size, vertical span) and its XZ-plane footprint, together with receptacle metadata indicating container properties and parent/child receptacle links. Predicate construction integrates image-space topology, depth ordering, and 3D geometry. The image-space layer includes *left of*, *right of*, *in front of*, *behind*, *overlapping*, and *near*, based on conservative thresholds over box centers, median depth within boxes, 2D overlap, and pixel-space proximity. The geometry layer activates *on*, *above*, and *below* when sufficient horizontal footprint overlap and vertical separation criteria are met, and assigns *in* using receptacle links reported by the simulator. All relations are emitted as duplicate-free ⟨subject, predicate, object⟩ triples drawn from the fixed vocabulary {*left of*, *right of*, *in front of*, *behind*, *overlapping*, *near*, *on*, *above*, *below*, *in*}.

The resulting multi-scene, multi-view dataset jointly supervises the detector with masks and bounding boxes under normalized category labels, and the predicate predictor with dense two-dimensional topology, depth-based front/behind cues, physical support and vertical ordering, and simulator-grounded containment. By combining complementary image-plane and metric 3D signals under randomized exploration, the generator produces training data that improves robustness and generalization of scene graph models under viewpoint changes, object articulation, and occlusions.

## C  THE USE OF LARGE LANGUAGE MODELS

Large language models (LLMs) play the following roles in this paper:

- The subject of research: We study the challenge of improving the decision-making performance of vision-language models.
- Writing assistant: We utilize LLMs to help proofread the manuscript and fix writing issues.

