# OpenReview forum: "Dive into the Scene: Autonomous Focus Plan Generation in Vision-Language Decision-Making"
_ICLR.cc/2026/Conference — Submitted to ICLR 2026_

### Official Review · Reviewer_H5pP · 2025-10-23

**Soundness:** 3
**Presentation:** 3
**Contribution:** 3
**Rating:** 6
**Confidence:** 2

**Summary:**

The authors introduce SceneDiver, a pipeline to generate a focus plan over visual scenes in order to aid decision making of VLMs. The VLM is tasked with constructing a coarse-to-fine focus plan which is used to modify the image by cropping or zooming into parts of the image. This is done to reduce the influence of distracting objects and other noise that could potentially (negatively) influence the decision making. The approach is evaluated on a robotic manipulation task and room navigation.

**Strengths:**

- important problem of reducing hallucination in visual decision making
- extensive evaluation across tasks and models

**Weaknesses:**

- no clear distinction from previous focus-plan methods. There is no benchmark comparison to Huang et al., 2024 and Zhu et al., 2025 which tackle the same problem. It should also be made clear, what the conceptual differences to those methods are.

**Questions:**

- Line 57: tasks-relevant -> task-relevant
- Line 194: missing brackets for citation

---

> ### Author Response · Authors · 2025-11-25
>
> $\textbf{Response to W: No clear distinction from previous focus-plan methods.}$  We acknowledge the reviewer's comment regarding the comparison with Huang et al. (2024) and Zhu et al. (2025). We would like to clarify the fundamental conceptual differences that distinguish our work from these approaches. The key distinction lies in the level of intervention: our method operates strictly at the input level (image editing), whereas the cited methods rely on model-internal modifications (parameter updates or internal activation adjustments).$\textbf{Zhu et al. (2025)}$ proposes Attention Calibration. Their method requires accessing and modifying the model's internal self-attention mechanism. Specifically, UAC computes a bias matrix to rectify attention weights during inference, and DAC requires a learnable plug-and-play module that is inserted into the transformer decoder layers. This necessitates "white-box" access to the model's architecture and internal states. $\textbf{Huang et al. (2024)}$ primarily focuses on benchmarking but proposes mitigating hallucinations via $\textbf{fine-tuning}$. Their approach involves unfreezing and retraining specific components, such as the vision encoder and connector, on generated datasets. This is a training-based approach that permanently alters the model's parameters.$\textbf{Our Approach:}$ In contrast, our method does $\textbf{not}$ require accessing the model's internal weights, attention maps, or performing any fine-tuning. We focus on $\textbf{guiding the model via image editing}$. By optimizing the visual input itself to highlight relevant regions, we treat the LVLM as a "black box."
>
> $\textbf{Response to Questions:}$ We thank the reviewer for this helpful suggestion.

---

### Official Review · Reviewer_Q8qL · 2025-10-26

**Soundness:** 3
**Presentation:** 2
**Contribution:** 3
**Rating:** 4
**Confidence:** 3

**Summary:**

This paper proposes SceneDiver, a novel method to mitigate visual hallucination in Vision-Language Models (VLMs) for decision-making tasks. SceneDiver generates a coarse-to-fine focus plan by having the VLM first reason over a scene graph and then zoom into local neighborhoods, creating an attention map that highlights task-critical objects. Experimental results in robotic manipulation and room navigation demonstrate that this approach significantly enhances decision-making performance and generalization ability.

**Strengths:**

1. The overall auto-focusing framework is well designed and makes sense.

2. The experiments cover robotic manipulation and navigation tasks.

**Weaknesses:**

1. Limited Experimental Scope and Baselines:
While the experiments in robotic manipulation and room navigation demonstrate the potential of SceneDiver, the experimental settings appear somewhat simplified and lack comparison against established, strong baselines. To convincingly validate the method's effectiveness and impact, it is crucial to benchmark it on widely-adopted and challenging benchmarks in the community, such as RLBench for manipulation or Habitat-Matterport3D for navigation. The absence of comparisons against state-of-the-art task-specific models (e.g., end-to-end IL/RL policies) or other VLM-based reasoning frameworks on these benchmarks makes it difficult to gauge the true performance leap offered by the proposed approach. Demonstrating superiority or competitive performance in these complex, standardized environments would significantly strengthen the paper's claims.

2. Architectural Disconnect with Mainstream Embodied Model:
The proposed method is presented as a "paradigm" for enhancing VLM-based decision-making, but it operates primarily as a pre-processing or intermediate reasoning step that modifies the visual input for a VLM. However, a dominant and successful trend in embodied AI is the use of end-to-end Vision-Language-Action (VLA) models, which directly map observations to actions. It is unclear how SceneDiver's two-stage, graph-based reasoning process can be integrated into such end-to-end architectures, which are often trained jointly for optimal performance. The authors should discuss the compatibility and potential integration strategies (e.g., using the focus plan as an intermediate tokenized representation) with these SOTA VLA models to address its applicability in the current research landscape.

**Questions:**

Here are some questions / minor weakness:

1. What is the computational and latency overhead introduced by the coarse-to-fine focus planning pipeline? Please provide an analysis of the inference time and computational footprint (e.g., FLOPs) of SceneDiver compared to the base VLM without focusing. A discussion on whether this overhead is justifiable given the performance gains is essential.

2. Can SceneDiver be integrated into existing navigation and manipulation frameworks (Both end-to-end VLA methods or hierarchical VLM-based methods are acceptable) in a general and plug-and-play manner?

---

> ### Author Response · Authors · 2025-11-25
>
> $\textbf{Response to W1: Experimental Scope and Baselines.}$ Our method is specifically designed to mitigate the multi-object hallucination issues encountered by VLMs during decision-making. Consequently, we selected robotic manipulation and room navigation tasks for evaluation, as these environments are populated with numerous distractors. Such settings are ideal for demonstrating the efficacy of our approach in filtering out irrelevant noise. We sincerely thank the reviewer for the suggestion and plan to incorporate more complex benchmarks in our future work to further broaden our evaluation.
>
> $\textbf{Response to W2 and Q2: Model Architecture and Future Integration.}$ We chose to build upon VLMs because, unlike VLA models which are often opaque black boxes, VLMs offer interpretability through Chain-of-Thought (CoT) reasoning, a capability that is currently challenging to replicate in VLA models. Our method exploits this by using scene graphs to augment the VLM's inference process for more reliable decision-making. Extending this interpretability and scene-graph-aware reasoning to VLA models is a key direction for our future research.
>
> $\textbf{Response to Q1：Computational Overhead and Latency.}$ We appreciate the reviewer’s scrutiny on efficiency. Indeed, the coarse-to-fine mechanism involves multiple VLM calls, leading to higher latency per step compared to single-pass methods.  However, we argue this is a favorable trade-off for two reasons. First, high-level reasoning in navigation does not require the high-frequency control needed for low-level stabilization. Second, and most importantly, SceneDiver significantly improves decision accuracy. By mitigating hallucinations, our robot avoids redundant backtracking and wrong turns. Thus, while the inference time per step increases, the total time to complete the task remains competitive due to fewer total actions required.

---

> > ### Comment · Reviewer_Q8qL · 2025-11-27
> >
> > Thanks for the authors' rebuttal. My major concern is about the Experimental Scope and Weak Baselines. These concerns are not solved yet, so I will remain my score. I recommend the authors to improve their experimental parts in a revised version.

---

### Official Review · Reviewer_X5bq · 2025-11-01

**Soundness:** 3
**Presentation:** 2
**Contribution:** 2
**Rating:** 4
**Confidence:** 3

**Summary:**

This paper proposes SceneDiver, a coarse-to-fine, two-stage focus plan generation pipeline aimed at mitigating object hallucination in Vision-Language Models (VLMs) during decision-making tasks. The method first constructs a scene graph and executes a virtual, coarse-grained plan to identify essential objects, followed by fine-grained, localized focusing around each anchor node. The resulting focus map is used to modify input images, thereby steering model attention to task-critical objects and suppressing distractors. Extensive experiments on robotic manipulation and room navigation benchmarks demonstrate improved decision-making performance and generalization. Ablation studies are performed to analyze the benefits of step-wise focusing and the robustness of the approach to scene graph quality.

**Strengths:**

- **Clear Problem Motivation:** The work targets the highly relevant issue of object hallucination in vision-language decision-making, specifically addressing scenarios where VLMs are used out-of-the-box without task-specific retraining.
- **Methodological Innovation:** SceneDiver’s two-stage, coarse-to-fine approach for generating a focus plan—comprising global scene graph planning followed by local fine-grained focusing—offers a structured strategy to mitigate distractor interference. This is visually clarified in Figure 1, which presents an end-to-end pipeline spanning instruction input to image modification.
- **Strong Empirical Support:** The paper provides comprehensive experiments (Tables 1 & 2) across multiple VLMs and tasks, showing consistent performance gains over baselines. Ablation analysis (Section 4.2, Table 2) convincingly dissects the contribution of each system component.
- **Thorough Visualizations:** The qualitative trajectories in Figures 3 and 4 and additional examples in the appendix (Figures 6–13) provide interpretability into the method’s effect on attention and decision steps.
- **Practical Robustness:** The paper demonstrates that the approach remains beneficial—even with untrained or partially trained scene graph generation—suggesting real-world applicability.
- **Reproducibility:** The methods and experimental setups are described in detail, including a stated intention to release code.

**Weaknesses:**

1. **Incomplete Positioning against Closest Prior Work:** Several recent and highly relevant works addressing hallucination in VLMs (see 'Potentially Missing Related Work') are not cited or discussed. This is a significant oversight given their strong alignment in motivation and technical foundation. Examples include works on visual inference chains, multi-view reasoning, and systematic hallucination benchmarking. This omission directly affects the paper’s contextualization and could impact the novelty claim.
2. **Math and Clarity Issues in Focus Map Construction (Section 3.4):** The mathematical exposition around score map construction and image editing suffers from a lack of clarity and rigor:
   - The definition of the coefficient $c(d)$ in Section 3.4.1 uses a Gaussian decay, but it is unclear how $\sigma$ is chosen, how sensitivity to hyperparameters is controlled, or how “distance to the image center” generalizes to complex scenes with off-center targets.
   - The blending equations for image emphasis/de-emphasis ($I' = I \odot (\beta + (1-\beta) s)$ and $I'' = (1-a) \odot I' + a \odot \mathcal{B}_{\sigma_b}(I)$) are correct in isolation, but the implementation details—such as the handling of per-object overlaps and thresholds—are not fully described. It is unclear how the method ensures that background suppression does not accidentally eliminate critical context in complex, occluded scenes. Furthermore, the method implicitly assumes $s$ is well-calibrated, yet there is no explicit calibration or adaptive thresholding, which can reduce robustness in cluttered or unfamiliar visual settings.
3. **Missing and/or Weak Baselines:** While ablation studies are included, some direct baselines from recent literature on hallucination reduction (such as multi-view and multi-hop reasoning or inference-chain frameworks) are missing. The lack of direct comparison to these approaches in Table 2 or the main results is a methodological gap that makes it difficult to situate SceneDiver’s relative merits and shortcomings.
4. **Over-reliance on Scene Graph Quality:** Despite some sensitivity analysis, the method fundamentally depends on the quality of object and relationship detection in the scene graph. While the results (discussed around Figure 5) show some robustness, there is insufficient quantitative exploration of failure modes—e.g., what happens if critical task-relevant objects are misdetected or missed in the initial coarse stage. The qualitative figures (e.g., Figure 4) focus on clean success cases rather than highlighting or dissecting difficult failures.
5. **Lack of Theoretical Guarantees or Deeper Analysis:** The paper motivates its pipeline with conceptual intuition but does not provide theoretical guarantees or analysis (e.g., convergence, error propagation, or attention calibration bounds). For example, in Section 3.3, the three focusing strategies are described, but no formalism is provided regarding their completeness or worst-case behavior. The stepwise approach may fail to recover in situations of cascading errors, which is not examined.
6. **Some Ambiguity in Experimental Protocols:** While the experimental setup is well detailed, it is not always clear how statistical significance is established in performance comparisons (Table 2)—variance is reported, but there’s no mention of hypothesis testing or statistical thresholds used to claim “significant enhancement.” Additionally, the supplementary visualizations (Figures 6–13) add interpretability but do not systematically evaluate failure cases.
7. **Potential Issues with Generalizability:** The method is tested on robotic manipulation and room navigation but may not generalize to visual domains with different object spatial relationships or more abstract task definitions (e.g., medical imaging or non-spatial language grounding). This limitation is briefly discussed but not empirically explored.
8. **Limited Discussion of Limitations in Main Text:** Most of the limitations and future work are condensed at the end, without being integrated into the problem discussion or experimental interpretation.

Potentially Missing Related Work

1. Zheng, H., Xu, T., Sun, H. (2024): *Thinking Before Looking: Improving Multimodal LLM Reasoning via Mitigating Visual Hallucination* — Proposes a Visual Inference Chain framework highly aligned in spirit with the focus on mitigating hallucinations in complex visual scenes. Should be cited and compared in Section 2 (Related Work, Object Hallucination) and included as a baseline/alternative strategy in comparisons (Tables 1 or 2).
2. Gu, Z., Chen, J., Liu, F. (2025): *MedVH: Toward Systematic Evaluation of Hallucination for Large Vision Language Models in the Medical Context* — Establishes systematic benchmarks for hallucination evaluation; relevant for broader claims and should be referenced in Section 4 (Experiments) when discussing generalizability and benchmarking.
3. Anonymous (2025): *See, Think, Hallucinate: Interpreting Reasoning and Hallucinations Beyond the First Hop in Vision-Language Models* — Investigates hallucinations beyond first-step reasoning, providing deep insights into multi-step or cascading errors similar to those in this work. Should be discussed in Sections 2 and 4, and ideally referenced alongside SceneDiver's multi-stage planning in Section 3.
4. Anonymous (2025): *Look, Compare, Decide: Alleviating Hallucination in Large Vision-Language Models via Multi-View Multi-Path Reasoning* — Introduces a training-free framework for hallucination reduction using multi-view, multi-path reasoning. Highly relevant as a direct baseline or discussed alternative; should be cited in Section 2 and quantitatively compared in Section 4/Table 2.
5. Anonymous (2025): *VIDHALLUC: Evaluating Temporal Hallucinations in Multimodal Large Language Models for Video Understanding* — Develops benchmarks for evaluating hallucinations in temporal/multimodal tasks. Should be referenced when situating SceneDiver’s applicability and generalization to temporally-evolving scenes.

**Questions:**

1. **Clarification on Focus Score Calibration:** How is the $\sigma$ parameter (Section 3.4.1) chosen for the attention map, and how sensitive are the results to its value? Is there an adaptive mechanism or heuristic for this, especially in cluttered or off-center tasks?
2. **Handling of Overlapping/Adjacent Candidates:** When multiple candidate objects overlap or are spatially adjacent, how is the focus map $A$ constructed to avoid unintended attention diffusion or masking of critical context?
3. **Failure Cases in Scene Graph Detection:** Can the authors provide additional quantitative analysis of how failures in scene graph extraction (missed objects/relations) propagate through the pipeline? What is the worst-case impact on downstream decision accuracy?
4. **Baselines Against Recent Hallucination-Mitigation Methods:** Are there technical reasons for not including Visual Inference Chain, Multi-View Reasoning, or related methods as direct baselines? If so, please clarify. If not, would the authors be able to add these for a fairer comparison?
5. **Statistical Significance:** What specific statistical tests or thresholds are used to determine whether the improvements in Table 2 are significant?

---

> ### Author Response · Authors · 2025-11-25
>
> $\textbf{Response to W1: Compare to Prior Work.}$  Our work specifically aims to mitigate $\textbf{multi-object hallucinations}$ in the context of decision-making tasks, rather than addressing hallucinations that occur during complex reasoning processes. Consequently, we find that the suggested literature falls outside the scope of our current study. Regarding the paper 'See, Think, Hallucinate: Interpreting Reasoning and Hallucinations Beyond the First Hop in Vision-Language Models', we are unable to include it as a citation as it is currently under submission to ICLR 2026 and is not yet a published reference.
>
> $\textbf{Response to W2 and Q1, 2: Math Clarity in Focus Map and Overlap Handling.}$  We clarify that the Gaussian coefficient $c(d)$ in $\textbf{Eq. (1)}$ is explicitly modeled after the "$\textbf{Center Bias}$"  mechanism in human visual attention , acting as a soft spatial prior. It does not enforce that objects must be centered; rather, it assigns higher attention weights to objects closer to the visual center $(x_c,y_c)$, simulating a natural gaze. Regarding implementation details and overlaps: for pixels shared by multiple objects in set $\mathcal{C}$, the score map $s$ is aggregated by prioritizing the object with the minimal distance $d_i$ (higher $c(d_i)$). This ensures that objects centrally located in the viewer's field of view naturally occlude or take precedence over peripheral ones. Furthermore, context preservation is guaranteed not just by the blending $\textbf{Eq. (2)-(4)}$, but primarily by the Scene Graph, which semantically filters relationships to define the candidate set $\mathcal{C}$ before map construction. This prevents the accidental suppression of critical context better than purely adaptive thresholding.
>
> $\textbf{Response to W3 and Q4: Missing Baselines.}$  Our work is specifically designed to mitigate multi-object hallucination within decision-making tasks, rather than addressing challenges such as multi-view reasoning. Consequently, the baselines we have selected are tailored to this specific objective. We believe these baselines provide a more appropriate and relevant comparison for evaluating our method's performance in decision-making scenarios.
>
> $\textbf{Response to W4 and Q3: Reliance on Scene Graph Quality and Failure Analysis.}$  Due to the limited perceptual and spatial reasoning capabilities of VLMs, we require an external tool to provide auxiliary information. While we acknowledge that scene graphs have certain limitations, they are sufficient for our task. Furthermore, to address the limitations of scene graphs, the proposed method exhibits inherent robustness through two mechanisms: the $\textbf{Focus Map}$ utilizes soft Gaussian weighting rather than hard masking to tolerate spatial noise, and the $\textbf{Scene Graph}$ provides semantic redundancy, enabling the LLM to leverage contextual relationships to infer plans even when individual detections are imperfect. We agree, however, that severe perception failures remain a limitation.
>
> $\textbf{Response to W5: Lack of Theoretical Guarantees.}$ Our method is specifically proposed to mitigate multi-object hallucinations in VLMs during decision-making. It is designed as a $\textbf{practical solution}$ tailored to real-world challenges, rather than a purely theoretical derivation. The effectiveness of our approach is substaintially validated by its strong performance on the benchmarks.
>
> $\textbf{Response to W6 and Q5: Experimental Protocols and Statistical Significance.}$ Regarding statistical significance, we reflect result stability by reporting the standard deviations across multiple experimental runs. Across all metrics, the consistent performance gains achieved by our method $\textbf{significantly exceed the observed variance}$, demonstrating the reliability of our approach. Regarding the visualizations in the Appendix, we acknowledge that the current presentation $\textbf{predominantly features}$ success cases. We recognize the high value of classifying and analyzing failure instances and consider this a priority for our future work. Specifically, we plan to systematically distinguish between perception errors and planning errors in subsequent research.
>
> $\textbf{Response to W7: Potential Issues with Generalizability.}$ We clarify that SceneDiver is explicitly tailored for $\textbf{embodied spatial planning}$ in realistic physical environments. The core mechanisms—specifically the reliance on physical object detectors and spatial relationship mining—are optimized for robotic manipulation and navigation tasks. Applying this to highly specialized domains like medical imaging or abstract non-spatial grounding would require fundamentally different upstream perception models and knowledge graphs, which falls outside the scope of this work. We consider the current evaluation on two distinct embodied domains sufficient to demonstrate the method’s effectiveness within its intended operational design.

---

### Official Review · Reviewer_qNoN · 2025-11-01

**Soundness:** 1
**Presentation:** 2
**Contribution:** 1
**Rating:** 2
**Confidence:** 3

**Summary:**

### Summary: Dive into the Scene

#### Research Problem
The paper addresses **visual hallucination** in Vision-Language Models (VLMs) when applied to complex decision-making tasks (like robot manipulation and navigation). In cluttered scenes, VLMs struggle to identify and focus on **task-relevant objects**, leading to poor decision accuracy.

#### Methodology (SceneDiver)
The authors propose **SceneDiver**, a novel, training-free, **coarse-to-fine, two-stage focus plan generation pipeline** to guide VLM attention:

1.  **Stage 1: Coarse-Grained Plan:** The VLM constructs a **scene graph** (object nodes and relationships) from the input image and executes a **virtual plan** over this graph. This process identifies coarse **anchor nodes** (critical objects) and minimizes the search space, validating consistency and mitigating multi-object hallucination.
2.  **Stage 2: Fine-Grained Focusing:** The VLM autonomously explores local neighborhoods around each anchor node, using simple strategies (Graph-Consistent Focus, Semantic-Guided Local Zoom, Stochastic Outward Search) to refine the focus and correct initial scene graph inaccuracies.
3.  **Result:** The generated focus plan is converted into a **Focus Score Map**, which is used to **edit (modify/dim)** the input image. This modification suppresses irrelevant background content and steers the VLM's attention toward essential targets, improving its grounded perception and decision-making.

#### Key Experiments
The approach is validated on challenging benchmarks:
* **Robotic Manipulation** (assembling pieces in cluttered scenes).
* **Room Navigation** (using complex scenes and hard-to-identify targets).

**Results** show that SceneDiver significantly **enhances the decision-making performance** and generalization ability of various off-the-shelf VLMs (e.g., Qwen, GPT-4o-mini, Gemini-2.5-Flash) by overcoming their perceptual limitations. Ablation studies confirm the superiority and robustness of the step-by-step focusing strategy.

**Strengths:**

### Strengths of the Paper

1.  **Effective Solution for VLM Visual Hallucination:** The paper directly addresses a critical and common failure mode of VLMs in complex real-world decision-making: **visual hallucination** and poor grounding due to cluttered scenes. By introducing an autonomous, training-free mechanism (SceneDiver) to generate a focus plan and edit the input image, the method successfully steers the VLM's attention to task-relevant objects, significantly improving decision accuracy.

2.  **Training-Free and Model-Agnostic Approach:** **SceneDiver** is implemented as a *training-free* pipeline using existing VLM reasoning capabilities (scene graph construction, virtual planning). This makes the approach highly practical, as it does not require collecting new paired data for training. Furthermore, it is **model-agnostic**, demonstrating performance gains when integrated with various off-the-shelf VLMs (e.g., Qwen, GPT-4o-mini, Gemini-2.5-Flash).

**Weaknesses:**

### Potential Weaknesses of the Paper

1.  **Dependency on Accurate Scene Graph Construction:** The Stage 1 coarse-grained plan hinges on the VLM's ability to accurately construct a **scene graph** (objects and relationships) from the initial image. In highly ambiguous, heavily occluded, or extremely cluttered scenes, an inaccurate initial scene graph will lead to the selection of incorrect **anchor nodes**, which may cause the subsequent fine-grained focusing stage to search the wrong area, resulting in failure.

2.  **Increased Computational Latency:** While the method is *training-free*, the **coarse-to-fine two-stage planning** (scene graph generation, virtual plan execution, and multiple local zooming/refinement steps) adds significant computational overhead. This sequential reasoning process increases the **inference latency** of the overall decision-making system, potentially limiting its deployment in time-critical, real-time applications.

3.  **Ambiguity in Image Editing (Focus Map):** The final step involves generating a **Focus Score Map** to **edit (modify/dim)** the input image. The effectiveness of this technique relies on the assumption that VLM attention mechanisms are reliably influenced by these simple visual edits. There is a risk that highly capable VLMs might "see through" the dimming or that the focus map might inadvertently obscure necessary contextual information.

4.  **Limited Scope of Visual Hallucination Addressed:** The method primarily tackles **multi-object hallucination** (distraction from irrelevant objects). It may not fully resolve other common types of visual hallucinations, such as **non-existent object hallucination** (the VLM imagining an object not present) or **attribute hallucination** (misidentifying an object's color or state), which are related to the VLM's internal generative biases rather than just focus.

**Questions:**

1. Although the method is training-free, could the author report the number of reasoning or action steps and the reasoning time taken by the relevant model when completing the task? This is beneficial for us to have a clear understanding of the computational complexity of the method.

2. Since the method tells us that this pipeline can overcome the visual illusion phenomenon, can the author further implement this method in the benchmarks of related manipulation and navigation tasks? For example, libero and habitat, calculate the completion rate of related tasks and provide relevant indicators?

---

> ### Author Response · Authors · 2025-11-25
>
> $\textbf{Response to W1：Dependency on Accurate Scene Graph Construction.}$ We would like to clarify a factual misunderstanding regarding our implementation. As detailed in $\textbf{Section 3.2}$, our Stage 1 utilizes the detection-based $\textbf{OvSGTR}$ model, $\textbf{not a VLM}$, to construct the scene graph. This design choice explicitly addresses the reviewer’s concern: unlike VLMs which may suffer from hallucinations in ambiguous scenes, OvSGTR anchors relationships on detected bounding boxes. Consequently, our method remains robust under the realistic clutter and partial occlusion scenarios present in the Room-Navigation Benchmark adopted in our evaluation, ensuring accurate anchor node selection without dependency on generative VLM capabilities.
>
> $\textbf{Response to W2 and Q1: Computational Latency and Complexity.}$ We thank the reviewer for the valuable feedback regarding computational overhead. In our current task context, we prioritize decision accuracy. We acknowledge that our method introduces additional inference latency; however, we consider this overhead acceptable. By mitigating hallucinations and preventing erroneous decisions, our approach significantly $\textbf{reduces the total number of action steps}$ required to complete a task. Consequently, despite the increased inference time per step, the total time-to-completion remains within a reasonable range. We fully agree with the reviewer’s suggestion and consider optimizing system efficiency a key direction for our future research.
>
> $\textbf{Response to W3: Focus Score Map Validity.}$ We appreciate the reviewer’s thoughtful comment on the Focus Score Map. However, our implementation in $\textbf{Section 3.4.2}$ is specifically designed to balance focal attention with global context. To prevent obscuring necessary scene information, $\textbf{Eq. (2)}$ utilizes a floor parameter beta $\beta$ to apply a "soft" mask, ensuring that background regions remain visible for global reasoning rather than being blacked out. Furthermore, to address the risk of the VLM "seeing through" the edits, $\textbf{Eq. (4)}$ applies Gaussian smoothing to the background. This operation physically suppresses high-frequency spatial features (edges and textures) in non-essential areas. Since Transformer attention mechanisms are inherently driven by such features, this "$\textbf{Blur Reverse Mask}$" strategy—validated as a highly effective visual prompt in recent literature (Yang et al., "Fine-Grained Visual Prompting", NeurIPS 2023)—effectively steers the model’s focus by altering the input's signal-to-noise ratio. This ensures the VLM attends to the sharp target regions without ignoring the broader scene layout or being misled by background noise.
>
> $\textbf{Response to W4: Scope of Hallucination.}$ We appreciate the reviewer's accurate observation that our method primarily targets multi-object hallucinations. Unlike broader studies on VLM hallucinations, we specifically focus on the impact of multi-object hallucinations on decision-making, a critical issue in robotic tasks. Although our method does not explicitly alter the VLM's internal generative biases, its design indirectly mitigates other types of hallucinations. Specifically, hallucinations of "$\textbf{non-existent objects}$" are largely suppressed by our first-stage detection-based scene graph (OvSGTR), as planning is grounded on physically detected bounding boxes rather than pure open-ended language generation. Furthermore, by utilizing the Focus Score Map to suppress background noise at the visual input level, we enhance the $\textbf{signal-to-noise ratio}$ of the target objects. This effectively reduces $\textbf{attribute misidentification}$ caused by visual clutter.
>
> $\textbf{Response to Q2: Generalizability on Other Benchmarks.}$ We appreciate the reviewer's valuable suggestion. While our method has already demonstrated strong performance on the two benchmarks reported in the paper, we are confident that it would also be effective on LIBERO and Habitat. However, due to the limited time constraints of the rebuttal period, we were unable to complete these additional evaluations. We will definitely consider implementing our method on broader benchmarks in the future.

---

### Author Response · Authors · 2025-11-25
**Author Response**

Dear Reviewers,

We would like to sincerely thank all the reviewers for the detailed reviews and suggestions. We have proposed point-to-point responses to the concerns and questions.

Best,

Authors

---

### Meta-Review · Area_Chair_MEEK · 2026-01-03

**Summary:**

The paper received mixed reviews, ranging from clear reject to borderline accept. Reviewers broadly agree that the problem is important and the empirical gains are real, but disagree on soundness, novelty positioning, and evaluation completeness.

**What the paper is trying to do (as reviewers understand it):**
- Address multi-object visual hallucination in VLM-based decision-making (robotic manipulation and navigation).
- Propose SceneDiver, a training-free, coarse-to-fine focus planning pipeline:
  - Stage 1: Detection-based scene graph + virtual planning to identify anchor objects.
  - Stage 2: Local neighborhood exploration to refine focus.
  - Final: A Focus Score Map edits the input image to steer VLM attention.

**Consensus strengths across reviewers:**
- Targets a real, well-known failure mode of VLMs in cluttered scenes.
- Training-free and model-agnostic, applicable to multiple off-the-shelf VLMs.
- Consistent performance gains across manipulation and navigation benchmarks.
- Clear qualitative visualizations and intuitive pipeline design.
- Robustness experiments showing tolerance to imperfect scene graphs.

**Core sources of disagreement:**
- Whether the method is sound and principled vs. a heuristic pipeline.
- Whether the evaluation is broad and competitive enough relative to recent hallucination-mitigation work.
- Whether the focus-map image editing is theoretically justified or fragile.
- Whether the paper is over-defensive in scope (rejecting related work comparisons).

**Reviewer Concerns:**

## Reviewer concerns that were addressed


**1. Dependency on scene graph quality:**
The method hinges on accurate scene graph construction; hallucinations in Stage 1 could cascade.

**Rebuttal response:**
- Clarified that Stage-1 scene graph is built using OvSGTR, a detection-based model, not a generative VLM.
- Emphasized grounding in detected bounding boxes, reducing hallucination risk.
- Argued robustness via:
  - Soft Gaussian weighting (not hard masking),
  - Semantic redundancy from graph relationships.


**2. Focus Score Map validity (image editing):**
Image dimming may be ignored by strong VLMs or obscure necessary context.

**Rebuttal response:**
- Explained soft masking with floor parameter β, preserving global context.
- Added Gaussian blur on background to suppress high-frequency features.
- Clarified overlap handling: prioritise the most central object when regions overlap.



**3. Computational overhead acknowledgment:**
Two-stage reasoning introduces latency.

**Rebuttal response:**
- Explicitly acknowledged increased per-step inference cost.
- Argued total task time decreases due to fewer wrong actions.


**4. Scope of hallucination claims:**
Method only handles some hallucination types.

**Rebuttal response:**
- Narrowed claim explicitly to **multi-object hallucination in decision-making**.
- Rejected broader hallucination benchmarks as out of scope.


**5. Math clarity in focus map construction:**
Equations unclear; overlap handling underspecified.

**Rebuttal response:**
- Explained Gaussian “center bias”.
- Detailed overlap resolution and aggregation rules.



---

## Reviewer concerns that remain partially or fully outstanding

**1. Missing comparisons to closely related hallucination-mitigation work:**
No comparison or discussion of visual inference chains or multi-view/multi-hop reasoning.

**Status:**
Still outstanding; weakens positioning.


**2. Lack of strong, standardized benchmarks:**
No LIBERO, RLBench, or Habitat-MP3D evaluations.

**Status:**
Unresolved.


**3. Absence of theoretical grounding:**
No guarantees or formal analysis.

**Status:**
Acknowledged but not resolved.


**4. Limited failure-case analysis:**
Focus on success cases.

**Status:**
Deferred to future work.


**5. Integration with end-to-end VLA models:**
Unclear compatibility with modern VLA architectures.

**Status:**
Unresolved.

---
## Overall

SceneDiver is a useful, training-free system that demonstrably improves VLM decision-making in cluttered scenes.
However, narrow evaluation, missing key baselines, and limited theoretical depth keep it just below a clear accept threshold.

**Reviewer Scores:**

Only reviewer H5pP seems positive. The other reviewers might not raise their scores enough to reach an "accept" level. The narrow evaluation and missing key baselines severely harm the paper quality.

---

### Decision · Program_Chairs · 2026-01-26

Reject